# Associations of the *CYP7A1* Gene Polymorphisms Located in the Promoter and Enhancer Regions with the Risk of Acute Coronary Syndrome, Plasma Cholesterol, and the Incidence of Diabetes

**DOI:** 10.3390/biomedicines12030617

**Published:** 2024-03-09

**Authors:** Gilberto Vargas-Alarcón, Óscar Pérez-Méndez, Rosalinda Posadas-Sánchez, Héctor González-Pacheco, María Luna-Luna, Galileo Escobedo, José Manuel Fragoso

**Affiliations:** 1Dirección de Investigación, Instituto Nacional de Cardiología Ignacio Chávez, Juan Badiano No. 1 Tlalpan, Mexico City 14080, Mexico; gvargas63@yahoo.com; 2Departamento de Biología Molecular, Instituto Nacional de Cardiología Ignacio Chávez, Juan Badiano No. 1, Tlalpan, Mexico City 14080, Mexico; opmendez@yahoo.com (Ó.P.-M.); mjluna.qfb@gmail.com (M.L.-L.); 3Departamento de Endocrinología, Instituto Nacional de Cardiología Ignacio Chávez, Juan Badiano No. 1 Tlalpan, Mexico City 14080, Mexico; rossy_posadas_s@yahoo.it; 4Unidad Coronaria, Instituto Nacional de Cardiología Ignacio Chávez, Juan Badiano No. 1 Tlalpan, Mexico City 14080, Mexico; hectorglezp@hotmail.com; 5Dirección de Investigación y Laboratorio de Inmunometabolismo, Hospital General de Mexico “Dr. Eduardo Liceaga”, Dr. Balmis No. 148, Cuauhtémoc, Mexico City 06720, Mexico; gescobedog@msn.com

**Keywords:** genetics, susceptibility, acute coronary syndrome, cholesterol 7 alpha-hydroxylase

## Abstract

Cholesterol-7-alpha hydroxylase (CYP7A1) is a key enzyme in the synthesis of bile salts, and its activity can contribute to determining cholesterol levels and, consequently, the risk of developing coronary atherosclerotic disease. We evaluated whether seven (rs3808607 *G/T*, rs9297994 *G/A*, rs10504255 *A/G*, rs8192870 *G/T*, rs2081687 *C/T*, rs1457043 *C/T,* and rs10107182 *C/T)* polymorphisms located in the promoter and enhancer regions of the *CYP7A1* gene, which have not been sufficiently explored, are candidates of risk markers of acute coronary syndrome (ACS) in the Mexican population. These polymorphisms were determined in a group of 1317 patients with ACS and 1046 control subjects. The results showed that, under different inheritance models, the alleles rs9297994 *G*, rs10504255 *G*, rs8192870 *T*, rs2081687 *T*, and rs10107182 *C* were significantly associated with an increased risk of ACS (*pC* < 0.05). In addition, the incidence of dyslipidemia among patients with ACS, notably high total cholesterol and LDL-cholesterol, and low HDL-cholesterol plasma levels, were more frequent in carriers of the same five risk alleles associated with ACS (*p* < 0.05). There was also an unexpected increased incidence of type 2 diabetes mellitus (T2DM) in patients with ACS who are homozygous for the rs2081687 *T*, rs9297944 *G*, rs10504255 *G*, and rs10107182 *C* alleles of the *CYP7A1* gene, suggesting that such gene variants enhance the development of coronary complications in patients with diabetes (*p* < 0.05). In summary, our study demonstrated that five polymorphisms situated in the promoter and enhancer regions of the *CYP7A1* gene are associated with the risk of ACS and higher incidences of dyslipidemia and T2DM in Mexican patients with ACS.

## 1. Introduction

Dyslipidemias play important roles as risk factors in the development of acute coronary syndrome (ACS), and they are the primary targets for current cardiovascular risk reduction strategies [1,2,3]. ACS is a multifactorial disease that results from the combination of background genetics and cardiovascular risk factors, such as obesity, hypertension, dyslipidemia, type 2 diabetes mellitus (T2DM), and smoking, among others, which play important roles in the development of the atherosclerotic plaque [1,2,3]. Cholesterol 7 alpha-hydroxylase (CYP7A1) is an enzyme that plays an important role in bile acid biosynthesis from cholesterol, a principal cholesterol removal pathway in the body [4,5,6]. In addition, after cloning the human gene [7,8], which is located in the region of q11-12 of chromosome 8, several studies have shown that CYP7A1 deficiency would reduce the conversion of cholesterol to bile acids, resulting in elevated cholesterol and LDL-cholesterol levels [5,9,10,11,12,13,14]. Accordingly, the CYP7A1 enzyme participates in cholesterol catabolism and bile acid homeostasis, which play important roles in the development of hypercholesterolemia and dyslipidemia [4,5,10]. In this context, recent studies have shown that some single nucleotide polymorphisms of the *CYP7A1* gene (rs3808607 *G/T*, rs9297994 *G/A*, rs10504255 *A/G*, rs8192870 *G/T*, rs2081687 *C/T*, rs1457043 *C/T*, and rs10107182 *C/T*) are associated with cardiovascular events such as coronary artery disease, T2DM, myocardial infarction, and hypercholesterolemia [5,9,10,11,12,13,14].

Based on these data, we propose that CYP7A1 gene polymorphisms could be associated with the prevalence of ACS through increased plasma lipid concentrations. In this context, we aimed to establish whether the rs3808607 *G/T*, rs9297994 *G/A*, rs10504255 *A/G*, rs8192870 *G/T*, rs2081687 *C/T*, rs1457043 *C/T*, and rs10107182 *C/T* polymorphisms are associated with the prevalence of ACS and plasma lipid concentrations.

## 2. Materials and Methods

### 2.1. Study Population

The sample size was calculated to one unmatched case and control study, using a power of 80% and an alpha of ≤0.05 as parameters. According to this, it was calculated that 528 individuals (264 patients with ACS and 264 control individuals) were required to carry out this study (Open Epi: version 3.01 (USA) http://www.openepi.com/SampleSize/SSCC.html (accessed on 17 June 2021)). This study included 2363 individuals (1317 patients with ACS and 1046 healthy controls). From July 2018 to November 2023, 1317 individuals (81% men and 19% women, with a mean age of 59.2 ± 10.8 years) were diagnosed with ACS at our institute. To diagnose ACS, we evaluated the patients’ clinical history; biomarkers of cardiac necrosis, such as the creatinine kinase isoenzymes and troponin I and II being above the upper limits of normal; as well as alterations in the electrocardiogram, in accordance with the guidelines of the European Society of Cardiology (ESC) and the American College of Cardiology (ACC) [15,16]. The control group included 1046 individuals (40.6% men and 59.3% women, with an average age of 51.2 ± 8.9 years) who were apparently healthy without a family history of premature CAD or atherosclerosis, according to the Mexican database of the Genetics of Atherosclerosis Disease (GEA) study [17]. Exclusion criteria included liver, kidney, thyroid, and oncological diseases, as well as congestive heart failure. In addition, all control subjects had to have a coronary calcium score equal to zero, as determined by computer tomography, to discard subclinical atherosclerosis [17]. The association between *CYP7A1* polymorphisms and lipid plasma levels was evaluated in the patients with ACS and healthy controls. In addition, all the subjects included in this study were ethnically matched, and only individuals who, for three generations, including their own, were born in Mexico were considered Mexican mestizos. This study was approved with project number 22-1288 by the Ethics and Research Commission of Instituto Nacional de Cardiología Ignacio Chávez, according to the Declaration of Helsinki guidelines. In addition, written informed consent was obtained from all individuals enrolled in this study.

### 2.2. Laboratory Analyses

Plasma concentrations of cholesterol and triglycerides were performed using enzymatic/colorimetric assays (Randox Laboratories, Crumlin, UK). High-density lipoprotein-cholesterol (HDL-C) concentrations were determined by the phosphotungstic acid-Mg^2+^ method. LDL-C concentrations were calculated using Friedewald’s formula in samples with triglyceride concentrations lower than 400 mg/dL [18]. Dyslipidemia was defined as increased plasma levels of one or more of the lipid profile parameters (total cholesterol > 200 mg/dL, LDL-C > 130 mg/dL, HDL-C < 40 mg/dL, or triglycerides > 150 mg/dL), according to the National Cholesterol Education Project Adult Treatment Panel III guidelines (NCEP-ATP III) (https://www.nhlbi.nih.gov/resources/third-report-expert-panel-detection-evaluation-and-treatment-high-blood-cholesterol-0 (accessed on 2 May 2023)). Type 2 diabetes mellitus (T2DM) was defined as fasting glucose levels ≥ 126 mg/dL [MSD manual guidelines professional version copyright © 2024 Merck & Co., Inc., Rahway, NJ, USA (https://www.msdmanuals.com/professional/endocrine-and-metabolic-disorders/diabetes-mellitus-and-disorders-of-carbohydrate-metabolism/diabetes-mellitus-dm#v29299021 (accessed on 2 May 2023))]. Systolic blood pressure ≥ 140 mmHg, diastolic blood pressure ≥ 90 mmHg, or the use of oral antihypertensive therapy were considered to establish hypertension presence.

### 2.3. Genetic Analysis

DNA was obtained from total blood samples [19]. The location of the rs3808607 *G/T*, rs9297994 *G/A*, rs10504255 *A/G*, rs8192870 *G/T*, rs2081687 *C/T*, rs1457043 *C/T*, and rs10107182 *C/T* polymorphisms included in this study are shown in Table 1. The determination of genotypes was performed using 5′exonuclease TaqMan assays on a QuantStudio 12K Flex Real-Time PCR system following the manufacturer’s instructions (Applied Biosystems, Foster City, CA, USA). As a validation method, 10% of the samples were genotyped twice; the results were concordant for all cases.

### 2.4. Statistical Analysis

The allele and genotype distribution of *CYP7A1* polymorphisms in patients with ACS and the controls were obtained by direct counting. The Hardy–Weinberg equilibrium was evaluated in both groups by the chi-squared test. Data analysis was performed with the SPSS program version 18.0 (IBM, Chicago, IL, USA). The Mann–Whitney U or Student’s *t*-test was used to compare the continuous variables. Categorical variables were analyzed with a chi-squared or Fisher’s exact test. The association between the rs3808607 *G/T*, rs9297994 *G/A*, rs10504255 *A/G*, rs8192870 *G/T*, rs2081687 *C/T*, rs1457043 *C/T,* and rs10107182 *C/T* SNPs and the prevalence of ACS was determined under the following inheritance models: the additive model, which compared the subgroup of individuals who were homozygotes for the major allele with the heterozygotes and the homozygotes carrying the minor allele; the codominant model that compared the homozygous individuals carrying the major allele with homozygotes of the minor allele; the dominant model, which compared the homozygous subjects carrying the major allele with the subgroup of individuals conformed by heterozygotes and homozygotes carrying the minor allele; the over-dominant (heterozygous) model that compared the subgroup of subjects who were homozygotes carrying the major allele and the homozygotes carrying the minor allele versus the heterozygous individuals; and the recessive model, which compared the subgroup of subjects who were heterozygotes or homozygous for the major allele versus homozygotes of the minor allele [20,21]. These analyses were adjusted for cardiovascular risk factors using logistic regression to determine whether the presence of the genetic variant was associated with the prevalence of the disease. The *p*-values were corrected by the Bonferroni test (*p*C) according to the number of comparisons per SNP in the different inheritance models. The values of *p* < 0.05 were considered statistically significant, and all odds ratios (ORs) were presented as 95% confidence intervals. Patients with ACS were grouped by genotypes and blood pressures, diabetes frequency, and glucose and plasma lipid concentrations (expressed as means ± SD) were compared by ANOVA tests and F-tests; *p* values < 0.05 were considered statistically significant. The statistical power of the association analyses between the *CYP7A1* gene polymorphisms and the incidence of ACS was set to 0.80 using the OpenEpi version 3.01 (USA) software available online [http://www.openepi.com/SampleSize/SSCC.htm (accessed on 17 June 2021)].

The design of haplotypes and the linkage disequilibrium analysis (LD, D’) were performed by means of the software Haploview, version 4.1 (Broad Institute of Massachusetts Institute of Technology and Harvard University, Cambridge, MA, USA). The database source of Haploview is the Human Haplotype Map project, which is used to determine the combination of alleles in a single gene, or alleles in multiple genes, along a chromosome that tends to be inherited together due to their proximity, providing statistical values of LD, D’, logarithm of the odds (LOD), and *r*^2^ [22].

## 3. Results

### 3.1. Characteristics of the Study Sample

The demographic, clinical, and biochemical parameters showed significant differences between patients with ACS and the controls (Table 2). As expected, the cardiovascular risk factors such as glucose plasma levels, frequency of hypertension, dyslipidemia, T2DM, and smoking habits were higher in patients than in the controls. Nonetheless, total cholesterol, triglycerides, and LDL-C plasma levels in patients with ACS were lower than those observed in the control group, probably due to the anti-dyslipidemic treatment and the change in lifestyle after the cardiac event.

### 3.2. Association between CYP7A1 Polymorphisms and ACS

Gene frequencies of the *CYP7A1* gene polymorphisms in ACS patients and the controls were in Hardy–Weinberg equilibrium. Only two polymorphisms (rs3808607 *G/T* and rs1457043 *C/T*) out of the seven polymorphisms studied in this work were not statistically different between patients with ACS and the controls. In contrast, the genotype frequencies of the rs9297994 *G/A*, rs10504255 *A/G*, rs8192870 *G/T*, rs2081687 *C/T*, and rs10107182 *C/T* SNPs showed significant differences between patients with ACS and healthy individuals (*p* < 0.05) (Appendix A). The association analysis of *CYP7A1* gene polymorphisms under different inheritance models showed that the *T* allele and the *TT* genotype of the rs2081687 *C/T* polymorphism were associated with a higher risk of developing ACS (OR = 1.36, 95% CI: 1.04–1.78, *pC*_Co-dominant_ = 0.022, OR = 1.41, 95% CI: 1.09–1.82, *pC*_Dominant_ = 0.009, OR = 1.33, 95% CI: 1.02–1.73, *pC*_Over-dominant_ = 0.036, and OR = 1.38, 95% CI: 1.10–1.74, *pC*_Additive_ = 0.006). In addition, the *G* allele and the *GG* genotype of the rs9297994 *G/A* SNP were associated with an increased risk of developing ACS (OR = 1.49, 95% CI: 1.14–1.94, *pC*_Co-dominant_ = 0.006, OR = 1.52, 95% CI: 1.17–1.97, *pC*_Dominant_ = 0.002, OR = 1.45, 95% CI: 1.11–1.90, *pC*_Over-dominant_ = 0.006, and OR = 1.46, 95% CI: 1.15–1.84, *pC*_Additive_ = 0.001). Similarly, the *C* allele and *CC* genotype of the rs10107182 *C/T* SNP were associated with a higher risk of developing ACS (OR = 2.37, 95% CI: 1.18–4.77, *pC*_Co-dominant_ = 0.034, OR = 2.30, 95% CI: 1.14–4.61, *pC*_Recessive_ = 0.016, and OR = 1.23, 95% CI: 1.01–1.50, *pC*_Additive_ = 0.038). Also, the *G* allele and *GG* genotype of the rs10504255 *A/G* SNP were associated with an increased risk of developing ACS (OR = 1.36, 95% CI: 1.05–1.76, *pC*_Co-dominant_ = 0.018, OR = 1.40, 95% CI: 1.09–1.80, *pC*_Dominant_ = 0.007, OR = 1.33, 95% CI: 1.03–1.71, *pC*_Over-dominant_ = 0.029, and OR = 1.37, 95% CI: 1.10–1.70, *pC*_Additive_ = 0.005). Finally, the *T* allele and *TT* genotype of the rs8192870 *G/T* polymorphism were associated with an increased risk of developing ACS (OR = 2.34, 95% CI: 1.16–4.72, *pC*_Co-dominant_ = 0.013, OR = 1.38, 95% CI: 1.07–1.78, *pC*_Dominant_ = 0.013, OR = 2.16, 95% CI: 1.07–4.33, *pC*_Recessive_ = 0.029, and OR = 1.38, 95% CI: 1.11–1.72, *pC*_Additive_ = 0.004) (Table 3). All models were adjusted for cardiovascular risk factors that were statistically different between groups, as indicated in Table 2, such as gender, age, dyslipidemia, systolic and diastolic pressures, BMI, glucose, total cholesterol, HDL-C, and smoking habit. 

### 3.3. Linkage Disequilibrium Analysis

The haplotype analysis was performed according to the location of the gene within the chromosome. In this context, four of the seven studied polymorphic sites (rs2081687 *C/T*, rs9297994 *G/A*, rs10107182 *C/T*, and rs10504255 *A/G*) of the enhancer region of the *CYP7A1* promoter showed a linkage disequilibrium (D′ > 0.85). This analysis revealed that the frequency of two (*CATA* and *TATA*) out of three haplotypes was significantly different between patients with ACS and the healthy controls (Table 4); the *CATA* haplotype was more frequent in the controls (84.8%) than in patients with ACS (77.5%) (*p* < 0.001), whereas the *TATA* haplotype was more frequent in patients with ACS (2.7%) that in the healthy controls (1.1%), suggesting that this haplotype is associated with a risk of developing ACS (*p* < 0.001). Also, the rs1457043 *C/T*, rs8192870 *G/T*, and rs3808607 *G/T* polymorphic sites at the promoter region were in linkage disequilibrium (D′ > 0.90). The frequencies of two haplotypes (*CGG* and *TTT*) were significantly different between the two groups (Table 4); the *CGG* haplotype was more common in the controls (8.9%) than in ACS patients (7.1%) (*p* = 0.023), while the *TTT* haplotype was more frequent in patients with ACS (2.0%) than in the controls (0.1%), suggesting that this haplotype is associated with an increased risk of developing ACS (*p* < 0.001).

### 3.4. Association between CYP7A1 Polymorphisms and Plasma Lipid Concentrations

Previous reports have suggested that the cholesterol 7 alpha-hydroxylase is associated with plasma lipid levels and with an increased risk of developing coronary artery disease and familial hypercholesterolemia [5,6,10]. In this context, we evaluated the possible functional effect of the rs9297994 *G/A*, rs10504255 *A/G*, rs8192870 *G/T*, rs2081687 *C/T*, and rs10107182 *C/T* SNPs by comparing the total cholesterol, LDL-C, HDL-C, triglyceride and glucose plasma concentrations, BMI, and systolic and diastolic pressures, in the patients with ACS grouped by their genotypes (Table 5). The rs2081687 *TT*, rs9297994 *GG*, and rs8192870 *TT* genotypes were associated with low HDL–C plasma concentrations (<40 mg/dL). In addition, the carriers of the rs10504255 *GG* and rs8192870 *TT* genotypes showed higher concentrations of total cholesterol and LDL–C (*p* < 0.05). Moreover, carriers of the rs9297994 *GG*, rs10504255 *GG*, rs8192870 *TT*, rs2081687 *TT*, and rs10107182 *CC* genotypes had increased systolic and diastolic blood pressures and glucose plasma concentrations (*p* < 0.05) (Table 5). On the other hand, the incidence of T2DM was significantly higher in the patients homozygous for the rs2081687 *TT*, rs9297994 *GG*, and rs10107182 *CC* genotypes that were also associated with ACS (Table 5). There was also a tendency for a higher incidence of T2DM in homozygous *GG* carriers of the rs10504255 *G/A* SNP, but the difference did not reach a statistical difference. In this context, glucose plasma levels were also significantly higher in the carriers of the rs2081687 *TT*, rs9297994 *GG*, rs10504255 *GG*, and rs10107182 *CC* genotypes (Table 5). However, we did not find any statistical difference in these parameters when the healthy control subjects were grouped by their corresponding genotypes (Appendix A). 

## 4. Discussion

In this study, we determined seven polymorphisms located in the *CYP7A1* gene that encode cholesterol 7 alpha-hydroxylase. This enzyme is the rate-limiting enzyme in the cholesterol conversion to bile acids [5,6,10]. Consequently, the relationship between *CYP7A1* gene polymorphisms and cardiovascular events has been explained mainly by LDL–C plasma levels. We found that five out of the seven of the studied polymorphisms in this work were associated with the risk of ACS. Previous studies in this field are scarce and controversial, and our work is one of the few studies that describe the association between these polymorphisms and ACS. In this context, we found that the rs2081687 *T* allele increased the risk of developing ACS. In line with this finding, several studies have shown that the rs2081687 *T* allele is associated with myocardial infarction [13] and with the risk of developing CAD [23,24,25]. In addition, a meta-analysis study showed that the rs2081687 *T* allele is associated with elevated LDL-C plasma levels [24]. Contrary to this evidence, we did not observe significant differences in LDLC plasma levels associated with the studied polymorphism in either the ACS patients or the control subjects. On the other hand, recent relevant data demonstrated that the rs8192870 *G/T* polymorphic site is related to metabolic disorders; the *T* allele has been reported to be associated with an increased risk of the development of T2DM in a Chinese population with CAD [14]. In our study, the *TT* genotype of the rs8192870 *G/T* polymorphic site was associated with a higher risk of ACS and elevated LDL-C plasma levels. Accordingly, in a Chinese population, the *TT* genotype was associated with a higher decrease in LDL-C plasma levels induced by atorvastatin than that observed in *GT* heterozygous and *GG* homozygous individuals [11]. Concerning the rs10504255 *A/G* genetic variant, the different alleles had similar regulation effects on gene expression when they were cloned into the reporter gene pGL4.23 [5]. Despite such evidence, the rs10504255 *G* allele was statistically associated with the incidence of ACS in our study. In addition, patients who were carriers of the rs10504255 *GG* genotype had an increased level of LDL-C in comparison with *AA* homozygous patients, whereas this difference was not present in the control group. Since the patients with ACS were treated with statins, these observations suggest that carriers of the *G* genotype had a lower response to statins as has been previously reported for other *CYP7A1* polymorphisms [26]. Therefore, the rs10504255 *A/G* genetic variant seems to be physiologically relevant and merits being specifically explored for pharmacogenomic purposes.

The information about a potential link between rs9297994 *G/A* and rs10107182 *C/T* SNPs with LDL–C plasma levels and cardiovascular diseases is scarce [5]. Nonetheless, data from genome-wide association studies showed that both polymorphisms located downstream of the *CYP7A1* enhancer region could impact LDL–C and total cholesterol plasma levels and that they were potentially associated with cardiovascular diseases [27,28,29,30]. In addition, Wang et al. reported that rs9297994 *G/A* and rs10107182 *C/T* SNPs seem to interact with the rs3808607 *G/T* polymorphic site to determine *CYP7A1* mRNA expression in human livers [5], but little is known about the effect of these genetic variants on the risk of developing coronary heart disease. In our study, we showed that the frequency of the rs9297994 *GG* and rs10107182 *CC* genotypes increased the risk of developing ACS; this association could be related to the previously reported high linkage disequilibrium (LD) between both SNPs [5]. However, there was not any apparent effect of rs9297994 *G/A* or rs10107182 *C/T* SNPs on LDL-C plasma levels. We consider that the lack of an association between the rs2081687 *T/C*, rs9297994 *G/A*, and rs10107182 *C/T* polymorphisms and LDL-C in patients with ACS may be due to statin intake. Consequently, the impact of *CYP7A1* gene polymorphisms on cholesterol plasma levels cannot be interpreted in our study. Nonetheless, we do not rule out an alternative mechanism that explains the relationship between these SNPs and an increased risk of ACS, i.e., the incidence of T2DM as discussed below.

Interestingly, we observed that the rs2081687 *TT*, rs9297994 *GG*, rs10107182 *CC*, and rs10504255 *GG* genotypes were associated with the incidence of T2DM and high plasma concentrations of glucose. In this context, it was demonstrated that a chronic insulin exposition induced the activation of the steroid regulatory element-binding protein-1c (SREBP-1c,) which in turn inhibits human *CYP7A1* gene transcription, thus regulating bile acid synthesis [31] (Figure 1). This evidence suggests that insulin resistance, which is characterized by increased levels of the hormone, may induce a decrease in bile acid synthesis, thus contributing to dyslipidemia in T2DM [32]. The regulation of insulin over bile acid synthesis seems to be bidirectional; Gerhard et al. found that diabetic patients had significantly higher serum bile acid plasma levels than healthy controls, which became even higher after diabetes remission in patients who underwent bariatric surgery [33]. In contrast, the administration of a bile acid sequestrant decreased glucose plasma concentrations and the percentage of glycated hemoglobin [31]. Despite a growing understanding of the interrelation between bile acids and glucose metabolism, the molecular bases behind their effects on diabetes remain unclear. In this context, our study strongly suggests that whichever the mechanisms are, the interaction between bile acids and glucose metabolism is dependent on *CYP7A1* gene variants (Figure 1). Such variants were also associated with higher LDL-C plasma levels in patients with ACS but not in the controls; it should be emphasized that the control group did not comprise subjects with diabetes in our study. Taken together, these observations indicate that *CYP7A1* variants enhance the risk of clinical manifestations of CAD in diabetic patients. We recognize that our study was not designed to analyze the interaction between *CYP7A1* gene polymorphisms and glucose metabolism but to determine the impact of poorly explored polymorphisms on ACS. Therefore, this study was not powered to perform more detailed statistical analyses with diabetes, but it sets a precedent to further explore the effect of *CYP7A1* gene polymorphisms on cardiovascular complications in diabetic patients.

Finally, we also found that the *TATA* and *TTT* haplotypes were associated with a high risk of developing ACS. It is important to consider that the frequency of these polymorphisms is dependent on the ethnic origin of the studied populations, and information about haplotypes is scarce in other populations. With this concept in mind, the frequency of the rs9297994 *G*, rs10504255 *G*, rs8192870 *T*, rs2081687 *T*, and rs10107182 *C* alleles in our population were 14.0%, 13.8%, 15.7%, 14.6%, and 14.0%, respectively (Appendix A). According to the data obtained from National Center for Biotechnology Information (https://www.ensembl.org/index.html accessed on 17 September 2023), the distribution of the rs9297994 *G*, rs10504255 *G*, rs8192870 *T*, rs2081687 *T*, and rs10107182 *C* alleles showed a high frequency in Mexican (from Los Angeles) (21.1%, 21.1%, 21.2%, 21.1%, and 21.1%, respectively), Asian (24.5%, 24.1%, 25.8%, 24.3%, and 24.5%, respectively), and Caucasian populations (36.8%, 36.5%, 37.0%, 36.5%, and 36.8%, respectively). On the other hand, the African people have a low frequency of the rs9297994 *G*, rs10504255 *G*, and rs10107182 *C* alleles (3.5%, 3.5%, and 2.7%, respectively), whereas the distribution of the rs8192870 *T* and rs2081687 *T* alleles (14.4% and 24.3%, respectively) were similar to Mexican mestizos and Asian populations (15.7% and 24.3%, respectively) (https://www.ensembl.org/index.html (accessed on 17 September 2023)). Taken together, our results and the different distribution of CYP7A1 gene polymorphisms in other populations, we propose that additional studies could help define the true role of these polymorphisms as universal markers of ACS susceptibility.

## 5. Conclusions

In summary, our study demonstrated that the rs9297994 *G/A*, rs10504255 *A/G*, rs8192870 *G/T*, rs2081687 *C/T*, and rs10107182 *C/T* polymorphisms of the *CYP7A1* gene are associated with the risk of developing ACS in a Mexican population. Also, it was possible to distinguish two haplotypes (*TATA* and *TTT*) associated with a high risk of developing ACS. On the other hand, the effect of the *CYP7A1* gene on LDL-C plasma levels was only apparent in patients with ACS but not in healthy controls. There was also an unexpected increase in the incidence of T2DM in the subgroup of patients with ACS who were homozygous for the rs2081687 *T*, rs9297944 *G*, rs10504255 *G*, and rs10107182 *C* alleles of the *CYP7A1* gene, suggesting that such gene variants enhance the development of coronary complications in diabetic patients. Further studies are warranted to elucidate the contribution of *CYP7A1* gene variants on the incidence of ACS, particularly in diabetic patients.

## Figures and Tables

**Figure 1 biomedicines-12-00617-f001:**
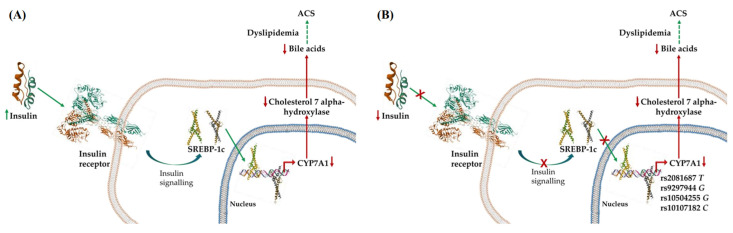
Simplified schematic representation of the relationship among diabetes, bile acids, and acute coronary syndrome (ACS), where CYP7A1 may be at the metabolic crossroads. (**A**) In normal metabolic conditions, insulin induces CYP7A1 repression through the activation of steroid-regulatory element-binding protein-1c (SREBP-1c). If insulin levels remain increased during prolonged periods, as in the insulin-resistance condition, decreased CYP7A1 expression may contribute to lower bile salt synthesis with the consequent increase in cholesterol plasma levels and the risk of ACS. (**B**) In the setting of diabetes, the minor alleles of the polymorphic sites located in the promoter and enhancer regions of the CYP7A1 gene contribute to maintaining a low expression of the gene, thus enhancing dyslipidemia and ACS development. Paradoxically, a pharmacological reduction in bile acids ameliorates insulin resistance (not indicated in the figure), suggesting a bidirectional metabolic control, i.e., insulin controlling bile acid expression and, vice versa, bile acids controlling insulin levels/effects by unknown mechanisms. Green arrows indicate a stimulatory effect and red arrows represent a downregulation of the pathway; red cross indicates that the effect is attenuated.

**Table 1 biomedicines-12-00617-t001:** Information of the studied polymorphism tested.

Gene Symbol	dbSNP ^a,b^	Chromosome	Chromosome Position	Change Base (pb)	Location in Gene
*CYP7A1*	rs2081687	8q11-12	58476006	*T > C*	Enhancer region/*CYP7A1* promoter
*CYP7A1*	rs9297994	8q11-12	58479765	*G > A*	Enhancer region/*CYP7A1* promoter
*CYP7A1*	rs10107182	8q11-12	58480178	*C > T*	Enhancer region/*CYP7A1* promoter
*CYP7A1*	rs10504255	8q11-12	58485902	*G > A*	Enhancer region/*CYP7A1* promoter
*CYP7A1*	rs1457043	8q11-12	58497880	*C > T*	Promoter region
*CYP7A1*	rs8192870	8q11-12	58499507	*T > G*	Promoter region
*CYP7A1*	rs3808607	8q11-12	58500365	*G > T*	Promoter region

^a^ SNP ID in database dbSNP; ^b^ Given name according to NCBI.

**Table 2 biomedicines-12-00617-t002:** Demographic, clinical, and biochemical characteristics of the study groups.

Characteristics		ACS Patients (*n* = 1317)	Healthy Controls (*n* = 1046)	*p*
Age (years)		59.2 ± 10.8	51.2 ± 8.9	<0.001
Gender *n* (%)	Male	1063 (80.7)	425 (40.6)	<0.001
	Female	254 (19.2)	621 (59.3)	
Hypertension, *n* (%)	Yes	740 (56)	199 (19)	<0.001
Type 2 diabetes mellitus, *n* (%)	Yes	835 (63)	109 (10)	<0.001
Dyslipidemia, *n* (%)	Yes	949 (72)	655 (63)	<0.001
Smoking, *n* (%)	Yes	635 (48)	232 (22)	<0.001
BMI (kg/m^2^)		27 [25–30]	28 [25–31]	0.061
Blood pressure (mmHg)	Systolic	130 [115–150]	112 [103–122]	<0.001
	Diastolic	80 [70–90]	70 [65–76]	<0.001
Glucose (mg/dL)		136 [109–201]	90 [84–97]	<0.001
Total cholesterol (mg/dL)		158 [126–190]	189 [167–211]	<0.001
HDL-C (mg/dL)		37 [31–44]	45 [36–55]	<0.001
LDL-C (mg/dL)		96 [70–126]	116 [95–134]	<0.001
Triglycerides (mg/dL)		141 [106–190]	145 [107–202]	0.171

Data are expressed as the mean ± SD or median [interquartile interval]. *p*-values were estimated using Student’s *t*-test or Mann–Whitney U test for continuous variables and the chi-square test for categorical values. ACS: acute coronary syndrome, BMI: body mass index.

**Table 3 biomedicines-12-00617-t003:** Association between *CYP7A1* polymorphisms and ACS in accordance with the inheritance models.

Polymorphic Site (rsID-Number)	Inheritance Model	Genotypes	ACS Patients *n* = 1317 *n*(%)	Controls *n* = 1046 *n*(%)	OR (95% CI)	*pC*
rs2081687	Co-dominant	*CC*	898 (0.682)	758 (0.725)		
		*CT*	367 (0.279)	269 (0.257)		
		*TT*	52 (0.040)	18 (0.017)	1.36 (1.04–1.78)	0.022
	Dominant	*CC*	898 (0.682)	758 (0.725)		
		*CT + TT*	419 (0.318)	287 (0.275)	1.41 (1.09–1.82)	0.009
	Recessive	*CC + CT*	1265 (0.960)	1027 (0.983)		
		*TT*	52 (0.040)	18 (0.017)	1.86 (0.86–4.05)	0.109
	Over-dominant	*CC + TT*	950 (0.721)	776 (0.743)		
		*CT*	367 (0.279)	269 (0.257)	1.33 (1.02–1.73)	0.036
	Additive	*-*	-	-	1.38 (1.10–1.74)	0.006
rs9297994	Co-dominant	*AA*	900 (0.683)	766 (0.733)		
		*AG*	366 (0.278)	264 (0.253)		
		*GG*	51 (0.039)	15 (0.014)	1.49 (1.14–1.94)	0.006
	Dominant	*AA*	900 (0.683)	766 (0.733)		
		*AG + GG*	417 (0.317)	279 (0.267)	1.52 (1.17–1.97)	0.002
	Recessive	*AA + AG*	1266 (0.961)	1030 (0.986)		
		*GG*	51 (0.039)	15 (0.014)	1.71 (0.76–3.82)	0.191
	Over-dominant	*AA + GG*	951 (0.722)	781 (0.747)		
		*AG*	366 (0.278)	264 (0.253)	1.45 (1.11–1.90)	0.006
	Additive	*-*	-	-	1.46 (1.15–1.84)	0.001
rs10107182	Co-dominant	*TT*	940 (0.714)	758 (0.733)		
		*TC*	336 (0.255)	261 (0.252)		
		*CC*	41 (0.031)	15 (0.014)	2.37 (1.18–4.77)	0.034
	Dominant	*TT*	940 (0.714)	758 (0.733)		
		*TC + CC*	377 (0.286)	276 (0.267)	1.19 (0.96–1.48)	0.119
	Recessive	*TT + TC*	1276 (0.997)	1019 (0.987)		
		*CC*	41 (0.031)	15 (0.014)	2.30 (1.14–4.61)	0.016
	Over-dominant	*TT + CC*	981 (0.745)	773 (0.748)		
		*TC*	336 (0.255)	261 (0.252)	1.09 (0.87–1.37)	0.429
	Additive	*-*	-	-	1.23 (1.01–1.50)	0.038
rs10504255	Co-dominant	*AA*	920 (0.699)	772 (0.740)		
		*AG*	353 (0.268)	254 (0.244)		
		*GG*	44 (0.033)	17 (0.016)	1.36 (1.05–1.76)	0.018
	Dominant	*AA*	920 (0.699)	772 (0.741)		
		*AG + GG*	397 (0.301)	271 (0.259)	1.40 (1.09–1.80)	0.007
	Recessive	*AA + AG*	1273 (0.967)	1026 (0.984)		
		*GG*	44 (0.033)	17 (0.016)	1.78 (0.87–3.65)	0.109
	Over-dominant	*AA + GG*	964 (0.732)	789 (0.757)		
		*AG*	353 (0.268)	254 (0.244)	1.33 (1.03–1.71)	0.029
	Additive	-	-	-	1.37 (1.10–1.70)	0.005
rs81922870	Co-dominant	*GG*	878 (0.667)	736 (0.705)		
		*GT*	382 (0.290)	288 (0.276)		
		*TT*	57 (0.043)	20 (0.019)	2.34 (1.16–4.72)	0.013
	Dominant	*GG*	878 (0.667)	736 (0.705)		
		*GT + TT*	439 (0.333)	308 (0.295)	1.38 (1.07–1.78)	0.013
	Recessive	*GG + GT*	1260 (0.957)	1024 (0.981)		
		*TT*	57 (0.043)	20 (0.019)	2.16 (1.07–4.33)	0.029
	Over-dominant	*GG + TT*	935 (0.710)	756 (0.724)		
		*GT*	382 (0.290)	288 (0.276)	1.26 (0.97–1.63)	0.083
	Additive	-	-	-	1.38 (1.11–1.72)	0.004

ACS: acute coronary syndrome, OR: odds ratio, CI: confidence interval, *pC*: corrected *p*-value. *p*-values were determined by logistic regression analysis, and ORs were adjusted for gender, age, dyslipidemia, systolic and diastolic pressures, body mass index, glucose, total cholesterol and HDL-C plasma concentrations, and smoking habit.

**Table 4 biomedicines-12-00617-t004:** Distribution of haplotypes between the rs2081687, rs9297994, rs10107182, and rs10504255 polymorphisms of the enhancer region, and the rs1457043, rs8192870, and rs3808607 SNPs of the promoter region in the study groups.

Polymorphic Site (rsID-Number)	ACS Patients *n* = 1314	Healthy Controls *n* = 1020	*p*
rs2081687 *T/C*—rs9297994 *G/A*—rs10107182 *C/T*—rs10504255 *G/A*	Hf (%)	Hf (%)	
*C A T A*	0.775	0.848	<0.001
*T G C G*	0.141	0.132	0.385
*T A T A*	0.027	0.011	<0.001
rs1457043 *C/T*—rs8192870 *T/G*—rs3808607 *G/T*	Hf (%)	Hf (%)	*p*
*T G T*	0.729	0.747	0.163
*C T G*	0.168	0.156	0.290
*C G G*	0.071	0.089	0.023
*T T T*	0.020	0.001	<0.001

Hf: haplotype frequency. The polymorphism order of haplotypes is according to the position in chromosome 8q11-12 [enhancer region (rs2081686—rs9297994—rs10107182—rs10504255) and promoter region (rs1457043—rs8192870—rs3808607)].

**Table 5 biomedicines-12-00617-t005:** Distribution of plasma lipid concentration according to the different genotypes of the rs2081687 *T/C*, rs8192870 *G/T*, rs9297994 *G/A*, rs10107182 *C/T*, and rs10504255 *A/G* polymorphisms in patients with ACS.

Gene/Parameters of Population	Genotypes			
*CYP7A1*	*rs2081687 T/C*			
	*CC* (*n* = 898)	*CT* (*n* = 367)	*TT* (*n* = 52)	*p **
Parameters				
BMI (kg/m^2^)	27.3 [25.5–30]	26.9 [24–29]	27.3 [25–29]	0.551
Blood pressure (mmHg)				
Systolic	130 [115–150]	129 [113–142]	132 [123–150]	0.037
Diastolic	80 [70–90]	80 [70–90]	81.5 [76.5–90]	0.041
Glucose (mg/dL)	137 [109–201]	131 [108–185]	160 [106–240]	0.038
T2DM (*n* = %)	537 (59.7)	206 (56.0)	37 (70.0)	0.037
Total cholesterol (mg/dL)	156 [126–188]	161 [124–196]	159.5 [129–197]	0.353
HDL-C (mg/dL)	37 [31–55]	37.6 [32–44]	35.6 [29–40]	0.041
LDL-C (mg/dL)	96 [70–125]	100 [71.6–130]	97.8 [71–125]	0.771
Triglycerides (mg/dL)	140 [106–189]	143 [107–193]	141 [128–219]	0.087
*CYP7A1*	*rs9297944 G/A*			
	*AA* (*n* = 900)	*AG* (*n* = 366)	*GG* (*n* = 51)	*p **
Parameters				
BMI (kg/m^2^)	27.3 [25–30]	27 [24.4–29]	27.3 [25–30]	0.548
Blood pressure (mmHg)				
Systolic	130 [116–150]	129 [112–146]	130 [123–148]	0.024
Diastolic	80 [70–90]	80 [70–90]	82 [78–90]	0.042
Glucose (mg/dL)	139 [109–205]	130 [107–183]	157 [107–227]	0.010
T2DM (*n* = %)	546 (60.0)	198 (54.0)	36 (70.5)	0.046
Total cholesterol (mg/dL)	156 [125–188]	160 [127–196]	158 [129–192]	0.511
HDL-C (mg/dL)	36.7 [31–44]	37.8 [32–45]	35.3 [30–39]	0.039
LDL-C (mg/dL)	96 [69–125]	98 [74–130]	97 [76–124]	0.570
Triglycerides (mg/dL)	142 [106–191]	139 [107–189]	135 [109–188]	0.775
*CYP7A1*	*rs10107182 C/T*			
	*TT* (*n* = 940)	*TC* (*n* = 336)	*CC* (*n* = 41)	*p **
Parameters				
BMI (kg/m^2^)	27.3 [25–30]	27 [24.5–30]	27 [25–28]	0.545
Blood pressure (mmHg)				
Systolic	130 [116–150]	129 [112–144]	128 [120–147]	0.044
Diastolic	80 [70–90]	80 [70–90]	83 [80–90]	0.042
Glucose (mg/dL)	137 [109–203]	130 [107–185]	160 [127–235]	0.022
T2DM (*n* = %)	553 (58.8)	180 (53.5)	31 (75.6)	0.010
Total cholesterol (mg/dL)	156 [126–188]	158 [124–195]	163 [137–192]	0.831
HDL-C (mg/dL)	36.8 [31–44]	37.5 [32–44]	36 [31–39]	0.679
LDL-C (mg/dL)	96 [70–124]	99 [71–130]	103 [81–129]	0.175
Triglycerides (mg/dL)	141 [106–191]	138 [107–186]	136 [122–213]	0.086
*CYP7A1*	*rs10504255 G/A*			
	*AA* (*n* = 920)	*AG* (*n* = 353)	*GG* (*n* = 44)	*p **
Parameters				
BMI (kg/m^2^)	27.3 [25–30]	27 [24.6–30]	27 [25–28]	0.682
Blood pressure (mmHg)				
Systolic	130 [115–150]	130 [113–146]	130 [124–148]	0.027
Diastolic	80 [70–90]	80 [70–90]	83 [79–90]	0.019
Glucose (mg/dL)	138 [109–203]	130 [107–186]	148 [107–226]	0.033
T2DM (*n* = %)	547 (59.4)	188 (53.2)	29 (65.9)	0.140
Total cholesterol (mg/dL)	156 [126–188]	158 [123–194]	166 [143–198]	0.030
HDL-C (mg/dL)	36.7 [31–44]	37.5 [32–44]	35.6 [29–39]	0.439
LDL-C (mg/dL)	96 [70–125]	96 [69–129]	106 [80–127]	0.035
Triglycerides (mg/dL)	142 [107–191]	138 [106–187]	135 [112–201]	0.373
*CYP7A1*	*rs81922870 T/G*			
	*GG (n = 878)*	*GT (n = 382)*	*TT (n = 57)*	*p **
Parameters				
BMI (kg/m^2^)	27 [25–30]	27 [25–30]	26.7 [24.5–28]	0.210
Blood pressure (mmHg)				
Systolic	130 [116–150]	129.5 [112–146]	128 [120–140]	0.033
Diastolic	80 [70–90]	80 [70–90]	80 [74–90]	0.015
Glucose (mg/dL)	137 [109–200]	133 [108–199.5]	134 [105–229]	0.554
T2DM (*n* = %)	518 (58.9)	215 (56.2)	35 (61.4)	0.314
Total cholesterol (mg/dL)	155 [124–188]	159 [129–192]	166 [137–206]	0.032
HDL-C (mg/dL)	36.8 [31–44]	37.5 [32.5–45]	36 [29–41]	0.035
LDL-C (mg/dL)	95.5 [67–126]	97 [76.5–124]	104 [76–144]	0.040
Triglycerides (mg/dL)	141 [106–190]	138 [107–192]	136 [112–197]	0.752

BMI: body mass index, HDL: high-density lipoprotein—cholesterol, LDL: low-density lipoprotein. Data are expressed as median [interquartile interval]. * ANOVA and F-test.

## Data Availability

The data shown in this work are available upon request from the corresponding author.

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
