# Peer review of "Associations of the CYP7A1 Gene Polymorphisms Located in the Promoter and Enhancer Regions with the Risk of Acute Coronary Syndrome, Plasma Cholesterol, and the Incidence of Diabetes"

_biomedicines, 2024, doi:10.3390/biomedicines12030617_

Round 1

Reviewer 1 Report

Comments and Suggestions for Authors

Dear Editor,

I really appreciate the paper from Vargas-Alarcón et al. Authors performed a well designed and scientifically correct research for identifying CYP7A1 Gene Polymorphisms as involved in ACS patients. The paper is well structured and the English is fluent. The scientific background is solid. 

Author Response

Thank you very much for your comments and suggestions to Manuscript-ID-biomedicines-2904126

We would like to thank the Reviewer for their comments; they have helped to improve the manuscript

Comments to the Author:

Reviewer #1: Comments and Suggestions for Authors

I really appreciate the paper from Vargas-Alarcón et al. Authors performed a well-designed and scientifically correct research for identifying CYP7A1 Gene Polymorphisms as involved in ACS patients. The paper is well structured and the English is fluent. The scientific background is solid. 

Answer: We really appreciated the comments of the reviewer. In this sense, we considered that our results more additional studies in other populations with different ethnic origin could help define the true role of these polymorphisms on the incidence of acute coronary syndrome, particularly in diabetic patients.

Reviewer 2 Report

Comments and Suggestions for Authors

This manuscript revealed the association of the CYP7A1 gene polymorphisms located in the promoter and enhancer region with risk of acute coronary syndrome (ACS), plasma cholesterol and incidence of diabetes. The results showed that five polymorphisms situated in promoter and enhancer region of the CYP7A1 gene were associated with the risk of ACS and higher incidence of dyslipidemia and T2DM in ACS Mexican patients. The topic is very interesting. The experimental design was well organized and the presentation was clear. It is worthy to be published after minor revision.

    It is quite difficult for readers to understand the complex mechanism under which the CYP7A1 gene polymorphisms is associated with the risk of diseases based on the literal descriptions. It will help the readers to understand it if the authors would provide action diagram to describe it, such as those in Lines 300-310.

Author Response

Thank you very much for your comments and suggestions to Manuscript ID Biomedicines-2904126

We would like to thank the Reviewer for their comments; they have helped to improve the manuscript

Comments to the Author:

Reviewer #2: Comments and Suggestions for Authors

This manuscript revealed the association of the CYP7A1 gene polymorphisms located in the promoter and enhancer region with risk of acute coronary syndrome (ACS), plasma cholesterol and incidence of diabetes. The results showed that five polymorphisms situated in promoter and enhancer region of the CYP7A1 gene were associated with the risk of ACS and higher incidence of dyslipidemia and T2DM in ACS Mexican patients. The topic is very interesting. The experimental design was well organized and the presentation was clear. It is worthy to be published after minor revision.

    It is quite difficult for readers to understand the complex mechanism under which the CYP7A1 gene polymorphisms is associated with the risk of diseases based on the literal descriptions. It will help the readers to understand it if the authors would provide action diagram to describe it, such as those in Lines 300-310.

Answer: Thank you for your constructive comment. The relationship between bile acids, diabetes and coronary heart disease is quite complex indeed, and it is still misunderstood. Following your kind suggestion, we have included Figure 1 that depicts a simplified schematic interpretation of such relationship, and its corresponding figure legend, page 11 of the corrected version of the manuscript. We hope that the figure makes clearer these concepts. 
